# Effects of Concentrate Feeding Sequence on Growth Performance, Nutrient Digestibility, VFA Production, and Fecal Microbiota of Weaned Donkeys

**DOI:** 10.3390/ani13182893

**Published:** 2023-09-12

**Authors:** Lan Xie, Jingya Xing, Xingzhen Qi, Ting Lu, Yaqian Jin, Muhammad Faheem Akhtar, Lanjie Li, Guiqin Liu

**Affiliations:** 1Shandong Engineering Technology Research Center for Efficient Breeding and Ecological Feeding of Black Donkey, Shandong Donkey Industry Technology Collaborative Innovation Center, College of Agronomy and Agricultural Engineering, Liaocheng University, Liaocheng 252000, China; m13210458205@163.com (L.X.); qxz17354603787@163.com (X.Q.); 19993443992@163.com (T.L.); jinyaqian@lcu.edu.cn (Y.J.); faheem_dear@hotmail.com (M.F.A.); 2College of Animal Science, Qingdao Agricultural University, Qingdao 266000, China; xingjingya1990@163.com; 3Office of International Programs, Liaocheng University, Liaocheng 252000, China

**Keywords:** weaned donkeys, acid-insoluble ash, nutrient digestibility, VFA, microbial composition

## Abstract

**Simple Summary:**

Feeding methods can influence animal feed intake and nutritional digestibility and boost economic benefits. Gut microbes are considered the “second genome” of the host, and different feeding methods can affect the dynamic balance of animal digestive tract microbes. Animal digestive tract microbes can also facilitate the utilization of indigestible nutrients by the body. As a result, we hypothesized that different concentrate feeding sequences would result in diverse gut microbial compositions, which could alter animal growth performance and nutrient digestibility indirectly. The Dezhou donkey (*Equus asinus*) is a good native herbivorous animal in China, with great foraging efficiency and productivity. There are currently few domestic reports on the best feeding methods for Dezhou donkeys. Therefore, this experiment was carried out to investigate the effects of concentrate feeding sequence on the growth performance, nutrient digestibility, volatile fatty acids (VFA), and fecal microbial composition of Dezhou donkeys (weaned) in order to provide new ideas for the selection of feeding methods for Dezhou donkeys in actual production.

**Abstract:**

In this study, effects on the growth performance, nutrient digestibility, volatile fatty acids (VFA) production, and fecal microbiota of weaned donkeys were observed using different concentrate feeding sequences. Fifteen healthy 6-month-old weaned male donkeys with a body weight of 117.13 ± 10.60 kg were randomly divided into three treatment groups, including group C1 (roughage-then-concentrate), group C2 (concentrate-then-roughage), and group C3 (total mixed ration, TMR). The experiment lasted 35 d. We measured nutrient digestion by the acid-insoluble ash method and analyzed the fecal microbiota of the weaned donkeys by high-throughput sequencing of 16s rRNA genes in the V_3_-V_4_ region. The results show that group C3 obtained the best growth performance, and the digestibility of crude protein (CP) and crude extract (EE) was significantly higher than that of group C1 (*p* < 0.05). Acetic acid, isobutyric acid, valeric acid, isovaleric acid, and caproic acid were notably different among all groups (*p* < 0.05). In addition, we observed that Firmicutes and Bacteroidetes were dominant in the fecal microbes of each group, and Firmicutes was significantly higher in group C3 (*p* < 0.05). At the genus level, the different genera were Treponema, Rikenellaceae-RC9-gut-group, Unidentified-F082, and Bacteroidales-RF16-group (*p* < 0.05). The prediction of fecal microbiota function by PICRUSt indicated that different feeding sequences had minimal impact on the function of the fecal microbiota, particularly on the high-abundance pathway. In summary, the concentrate feeding sequence changed the composition of the fecal microbe of weaned donkeys.

## 1. Introduction

Conventional feeding for most herbivores involves segregating the concentrate and coarse feed. The utilization of this feeding method is associated with certain drawbacks, including decreased feed intake and an inefficient use of feed resources [1]. It does not promote the development of large-scale intensive management. Therefore, we sought to explore the potential effects of altering the feeding sequence. Total mixed ration (TMR) refers to a diet that provides a generally balanced amount of nutrients based on the dietary requirements of animals at different growth stages by thoroughly mixing various additives, concentrate feeds with varying nutrient levels, and roughage feeds in a particular proportion [2]. TMR feeding is a standardized, large-scale technology commonly used for ruminants [3,4]. Previous research demonstrated that different feeding methods could change the feed intake of dairy horses, influence the body’s ability to absorb nutrients, and boost economic rewards [5]. Furthermore, TMR feeding has been shown to increase the daily gain of Sika deer, enhance rumen fermentation, increase the pace at which dietary nutrients are utilized, and improve Sika deer’s metabolism of protein [6].

Donkeys, known as herbivores, degrade ingested structural carbohydrates via bacteria in their gut to supply nutrients for their growth [7,8]. An intricate and mutually adapted micro-ecosystem is formed by the host and their gut microorganisms through coevolution, which is essential for the maintenance and optimal physiological function of the gut [9,10]. If the homeostasis balance of the gut microbiome is interrupted, it can bring about gut diseases, dietary changes, and even death. In addition to its role in digestion, there is evidence that the gut microbiota play a part in efficient nutrient utilization, the development of the digestive tract, immunity, and host health [11,12,13]. In addition, volatile fatty acids produced by microbial digestion of dietary fiber provides a large portion of a horse’s daily energy requirements [14]. Fecal metabolites can reflect the digestion and absorption of nutrients by gut bacteria and the host gastrointestinal tract, as well as better explain the influence of host–microbiome and metabolome interactions on growth performance [15]. Zarrinpar showed that different feeding methods can modify the structure and abundance of the gut microbiota in mice, gut microbes may regulate mice’s energy metabolism [16]. Feeding methods may disrupt the microbial homeostasis in the digestive tract of animals [17].

Based on the above findings, it is hypothesized that the concentrate feeding sequence affects the gut microbiome, indirectly affecting the animal’s growth performance and nutrient digestibility. Donkey breeding is currently receiving increasing attention in China. But compared to other livestock, the donkey sector needs nutritional guidelines and feeding standards appropriate for their unique physiology. In particular, there are few domestic reports on the most suitable feeding methods for weaned donkeys. Therefore, the aim of this experiment was to explore the effects of concentrate feeding sequences on the growth performance, nutrient digestibility, VFA production, and fecal microflora of weaned donkeys. The findings will serve as a valuable reference for selecting the appropriate feeding methods in practical production.

## 2. Materials and Methods

### 2.1. Animals, Diets, and Feeding

Nutritional assessment was picked up before the test under the guidance of Cappai et al. [18]. Fifteen healthy 6-month-old weaned male donkeys with a body weight of 117.13 ± 10.60 kg were randomly divided into three groups: group C1 (roughage-then-concentrate), group C2 (concentrate-then-roughage), and group C3 (total mixed ration, TMR), with 5 donkeys in each group, which were provided by Dong-E-E-Jiao Co., Ltd. (Liaocheng, Shandong, China) The whole experiment was carried out at a breeding facility for black donkeys in Liaocheng. The primary ingredients used to make the concentrate feed for the trial diet were corn grain, wheat flour, and corn germ meal, and the composition and nutrient contents of the concentrate feed are listed in Table 1. The roughage was peanut vine, and its nutritional content is listed in Table 2. The experimental donkeys were raised in a single house and given concentrate (1.3% of their body weight) and roughage (the ratio of roughage to concentrate was 3/2) at 8:00 and 17:00 daily. The amount of feeding was weighed and adjusted in the middle of the experiment. Group C1 was fed roughage for half an hour, and then concentrated feed, and vice versa for group C2. Group C3 had a mixture of roughage, concentrate, and water in equal proportions. The experiment lasted 35 d (from 3 September to 8 October 2022, temperature of 23–34 °C), and the pre-feeding period was 7 d. The donkey pens were routinely cleaned and disinfected during the trial, and they drank water freely. All weaned donkeys remained clinically healthy through veterinary examinations during the experiment period.

### 2.2. Major Reagents, Instruments

The reagents used were 10% sulfuric acid, 40% sodium hydroxide (NaOH), 2% boric acid (H_3_BO_3_), 15% phosphoric acid (H_3_PO_4_), 4-methylvaleric acid, petroleum ether (boiling range of 40–60 °C), and 3 mol/L hydrochloric acid (HCl). All the above reagents were chemically pure and obtained from the Sinopharm Chemical Reagent Company (Ji’nan, China).

The instruments used were an automatic Kjeldahl nitrogen analyzer (Hanon, K9840, Ji’nan, China), a fat analyzer (Hanon, SOX406, Ji’nan, China), a graphite digester (Hanon, SH220F, Ji’nan, China), an electronic balance (OHAUS, PX124ZH/E, Shanghai, China), a fiber analyzer (Ringbio, R-200, Ji’nan, China), a refrigerated centrifuge (cence, H2050-R, Hunan China), and a Thermo ISQ 7000 mass spectrometer (Thermo Fisher Scientific, Waltham, MA, USA).

### 2.3. Samples Collection

Five days prior to the completion of the experiment, the feces were continuously collected, weighed, and then added to a 10% sulfuric acid solution at a ratio of 5% based on their fresh weight. The mixture was subsequently stored at −20 °C for future use. After mixing all the feces, a sample (200 g) was randomly taken and dried to constant weight at 65 °C. Finally, the samples were crushed through a 40-mesh screen to produce air-dried samples for laboratory analysis. At the end of the experiment, the rectal feces of weaned donkeys in each group were collected and placed in a sterile enzyme-free centrifuge tube and frozen at −80 °C.

### 2.4. Growth Performance, Nutrient Digestibility, and VFA

The donkeys were weighed the morning before and at the end of the experiment. To calculate the average daily increase in weight of each donkey, the initial body weight was subtracted from the final body weight and divided by the days of the experiment. The residual material was weighed daily while the grass-weighing material was fed into the hay bucket every five days to determine the average daily feed intake. The average daily gain and daily feed intake were used to calculate the feed-to-gain ratio. 

The contents of crude protein (CP), crude extract (EE), crude fiber (CF), acid detergent fiber (ADF), and neutral detergent fiber (NDF) in the concentrate feed and peanut vine were determined. The crude protein content was determined using the Kjeldahl nitrogen determination method, the crude fat content was determined using the Soxhlet extraction method, and the contents of crude fiber, acid detergent fiber, and neutral detergent fiber were analyzed using a gravimetric method. According to the AOAC methods [19,20], the samples were examined for the digestibility of CP, EE, CF, NDF, and ADF.

Then, 50 mg samples were put into a 2 mL centrifuge tube, 15% phosphoric acid (50 μL) was added, and then 125 μg/mL internal standard (4-methylvaleric acid) solution (100 μL) and ether (400 μL) homogenate were added and mixed for 1 min. Next, the samples were centrifuged at 12,000× *g* at 4 °C for 10 min, and finally, the supernatant was put into a vial prior to GC-MS analysis [21].

### 2.5. Genomic DNA Extraction, PCR Amplification and Purification, Sequencing

The total genome DNA was extracted using the CTAB/SDS method. The DNA concentration and purity were monitored using 1% agarose gels. Next, the DNA was diluted to a concentration of 1 ng/uL using sterile water, depending on its initial concentration. The diluted genomic DNA was utilized as the template, and specific primers (F: CCTAYGGGRBGCASCAG; R: GGACTACNNGGGTATCTAAT) with barcodes were selected for PCR amplification of the target region using a Phusion® High-Fidelity PCR Master Mix (M0532S, New England Biolabs, Inc., Ipswich, MA, USA) and a high-efficiency, high-fidelity enzyme. After that, a NEBNext® Ultra™ IIDNA Library Prep Kit (Cat No. 5, New England Biolabs, Inc., Ipswich, Catalog #:E7645B, USA) was used to construct the library, and the constructed library was quantified by Qubit and Q-PCR. Finally, the library was sequenced on an Illumina NovaSeg6000 (Illumina, San Diego, CA, USA) platform.

### 2.6. Bioinformatics Analysis

To ensure the accuracy of the subsequent analysis, fastp (Version 0.20.0) software was used to quality filter the raw tags to produce high-quality clean tags, followed by the use of Vsearch (Version 2.15.0) to find and remove any chimera sequences to produce effective tags [22]. The alpha diversity indices (Observed-OTUs, Chao1, Shannon, Simpson) were evaluated using QIIME2. To evaluate the complexity of the community composition and differences among the groups, the beta diversity was estimated by QIIME2 based on weighted-unifrac distances. Cluster analysis was performed using principal coordinates analysis (PCoA), which reduced the dimensionality of the original variables using the ade4 and ggplot2 packages in R software (Version 3.5.3). Then, the adonis and anosim functions in QIIME2 software (Version 202006) were employed to analyze the significant differences in the community structure among the groups. Finally, species analysis with significant differences among the groups was performed using LEfSe or R software. LEfSe analysis was performed using LEfSe software. The default LDA score threshold is 4. Additionally, PICRUSt2 software (Version 2.1.2-b) was employed for functional analysis in order to investigate the functionality of the bacterial communities. We regarded *p* < 0.05 as indicating significant differences among the groups.

### 2.7. Statistical Analysis

Data analysis was performed using a one-way analysis of variance ANOVA procedure with SPSS software version 17.0. The post hoc test was determined using Tukey HSD tests. The data are expressed as the mean ± standard error. *p* ≤ 0.05 indicates statistical significance.

## 3. Results

### 3.1. Effects of Different Concentrate Feeding Sequences on Growth Performance

The data on the growth performance are shown in Table 3. The initial body weight (BW) was similar in all three groups, with group C3 having the highest final BW. The differences among the three groups were not statistically significant (*p* > 0.05), but the average daily gain (ADG) was significantly different (*p* < 0.05). The average daily feed intake (ADFI) of group C3 was significantly higher than that of group C1 (*p* < 0.05). The feed intake/gain (F/G) in group C3 was significantly lower than that in group C1 (*p* < 0.05).

### 3.2. Effects of Different Concentrate Feeding Sequences on Apparent Digestibility

According to Table 4, the apparent digestibility of CP and EE in group C1 was significantly lower than in the other groups (*p* < 0.05). There were no significant differences in the apparent digestibility of CF, NDF, and ADF among any groups (*p* > 0.05).

### 3.3. Effect of Different Concentrate Feeding Sequences on VFA Concentration

Table 5 shows significant differences in the total volatile fatty acids among the three groups (*p* < 0.05). Acetic acid, isobutyric acid, valerate acid, and isovalerate acid were significantly different (*p* < 0.05), and there were no significant differences in the propionic acid and butyric acid contents among any of the groups (*p* > 0.05).

### 3.4. Effects of Different Concentrate Feeding Sequences on Intestinal Microflora Composition

#### 3.4.1. Analysis of α-Diversity of Species

In order to investigate the effect of the concentrate feeding sequence on the fecal microbiota composition in weaned donkeys, the fecal microbiota were quantified using 16S rRNA sequencing. The number of high-quality sequences obtained in total samples was 1,226,185. Moreover, the general 16S rRNA operational taxonomic units (OTUs) reached 4686 based on 97% sequence similarity. The Shannon index in group C1 was significantly higher than that in group C2 (*p* < 0.05), and the OTU and Chao1 index in group C1 were significantly higher than those in the other groups (*p* < 0.05). There was no statistically significant difference in the Simpson index among the sequences of all groups (*p* > 0.05) (Table 6).

#### 3.4.2. Relative Abundance of Microbial Species at Phylum and Genus Levels

The top 10 phyla with relative abundance after the effective sequences from each group were clustered and annotated at the phylum and genus levels (Figure 1a). The top 10 species in terms of relative abundance at the genus level are displayed in Figure 1b.

At the phylum level (Figure 1a), the relative abundance of Bacteroidota (C1, 40.87%; C2, 53.22%; C3, 49.44%) was significantly different among the three groups (*p* < 0.05), the relative abundance of Firmicutes in group C1 (39.08%) was significantly higher than that in the other groups (C2, 30.25%; C3, 29.05%) (*p* < 0.05), and the relative abundance of Spirochaetota in group C3 (8.25%) was significantly higher than the other groups (C1, 3.62%; C2, 3.88%) (*p* < 0.05). There were no significant differences in the levels of Myxococcota, Chloroflexi, Verrucomicrobiota, Acidobacteriota, Actinobacteriota, Proteobacteria, or Fibrobacterota (*p* > 0.05).

Around 50% of the bacteria species found were undefined (others). According to the identified bacterial genera (Figure 1b), the Rikenellaceae-RC9-gut-group of group C2 (24.49%) was considerably higher than that of the other groups (C1, 17.50%; C3, 19.11%) (*p* < 0.05); the Unidentified-F082 of group C1 (7.68%) was considerably lower than that of the other groups (C2, 11.63%; C3, 12.65%) (*p* < 0.05); and Treponema in group C3 (6.55%) was quite higher than that in the other two groups (C1, 2.57%; C2, 3.21%) (*p* < 0.05). There was no significant difference in the remaining seven genera (*p* > 0.05).

#### 3.4.3. Microbial Community Analysis of Fecal Microbiota

PCoA demonstrated that the microbial communities from the fecal differ across various concentrate feeding sequence (Figure 2a). According to the weighted unifrac distance analysis, the community composition of the fecal at group C1 was substantially different from that of other groups (C2 and C3 groups) (AMOVA < 0.05); PC1 explained 37.15% of the difference among the groups. Linear discriminant analysis (LDA) and effect size (LEfSe) analysis showed that the enriched microbial community was Bacteroidota in group C2. When comparing various bacteria in the C1 and C3 groups, there were proportional differences among Firmicutes, Clostridia, and Oscillospirales in group C1, and between Spirochaetota and Prevotellaceae_UCG_004 in group C3 (Figure 2b). Cladograms were constructed to depict the phylogenetic distribution of the distinct bacteria in the three groups (Figure 2c).

#### 3.4.4. Functional Predictions of the Fecal Microbiota Using PICRUSt

We predicted the function of the fecal microbiota of weaned donkeys at various concentrate feeding sequences using PICRUSt. Level 1 included the most represented metabolic pathways of the fecal bacterial flora in weaned donkeys. In level 2, the mainly pathways were amino acid metabolism, carbohydrate metabolism, membrane transport, and replication and repair. However, there were no significant differences in these pathways among the treatments (*p* > 0.05) (Figure 3a,b). In the level 1 KEGG, the relative abundances of metabolism, genetic information processing, and environmental information processing were at the highest levels in different concentrate feeding sequences (Figure 3c). In the level 2 KEGG, the relative abundances of amino acid metabolism, membrane transport, and carbohydrate metabolism were the highest in all groups and showed no significant differences. The cell motility pathway was significantly lower in group C2 than in the other groups (*p* < 0.05), while the replication and repair, immune system, and translation pathways were all significantly lower in group C1 compared to the other groups (*p* < 0.05) (Figure 3d).

## 4. Discussion

Previous studies have found that feeding methods could influence animal feeding behavior and nutrient metabolism, impacting how much protein and fat is deposited [23,24]. However, only a handful of studies have described the effects of various concentrate feeding sequences on the growth performance of donkeys. In this experiment, total mixed ration feeding significantly increased the ADG and decreased the F/G of donkeys. It was discovered that the growth performance of group C3 was better than that of group C1, although the difference was not statistically significant, which may be related to the feeding cycle’s brief duration. Liu Mingli [25] discovered that group C2 had the best growth performance, which is inconsistent with the results of this study. This difference may be caused by the different growth stages of the experimental animals. Moore-Colyer et al.’s study [26] showed that high growth rates can be achieved by Thoroughbred foals when fed a total mixed fiber ration (TMFR), and these growth rates were comparable to those achieved when a conventional cereal-based creep feed was fed. A TMFR can maintain a healthy gut environment by raising the pH and reducing lactate generation. Liu Fenghua’s experiment indicates that a TMR pellet feeding mode can improve the average daily gain of meat donkey foals [27]; this is inconsistent with this study, possibly because we used conventional TMR, not TMR pellet feed.

Animals’ ability to utilize nutrients in feed is an essential indicator of their nutritional value [28]. Feed fermentation can decrease the intestinal pH value because a large amount of volatile fatty acids is produced in the fermentation process [29]. Therefore, we speculate that the decrease in pH will affect the digestibility of nutrients. Jouany’s study showed that the apparent digestibility of CP in horse high-starch diets was greater than in high-fiber diets, independently of live yeast culture supplementation [30]. In fact, not all of the starch in a high-cereal grain diet could be converted into energy. If the starch content is high, it may surpass the horse intestine’s capacity to digest it and result in a high glycemic reaction [31]. Our study found that the dietary nutrients’ digestibility (CP and EE) in group C1 was lower than that in the other two groups, which was caused by the feeding sequence of concentrate, and may also be related to the retention time of concentrate (which is rich in starch) in the gastrointestinal tract. However, the effect of the retention time on gut starch digestibility is arguable. Some of the literature shows an increase in digestibility with prolonged retention time [32,33], while the de Fombelle discovered no interaction between time and digestibility [34]. Therefore, the relationship between the concentrate retention time in donkey gut and the concentrate digestibility needs to be further studied.

Dietary fiber is not digested by animal digestive enzymes [35], but microbes in the hindgut ferment complex carbohydrates in fiber-rich diets into VFAs, contributing 60–70% of the horse’s daily energy requirements [36]. The predominant VFAs are acetic, propionic, and butyric acid [37]. Acetic acid releases ATP to provide energy for the body through the tricarboxylic acid cycle (TCA cycle) process, while propionic acid serves as a precursor to the creation of glucose. Animals can obtain more energy for development when propionic acid concentrations are more significant [38]. This study found that group C3, with better growth performance had less propionic acid content, but the difference was not significant. The inconsistent result is most likely caused by the fact that fecal microorganisms cannot completely replace gut microbes. Butyric acid is a source of metabolic energy for intestinal cells, has anti-inflammatory properties, helps maintain the integrity of the host’s mucosal barrier, and is associated with regulating immune response, contributing to maintaining intestinal microbial equilibrium [39]. Butyrate may mediate the effects of diet and gut microbiota on host appetite, metabolism, and adiposity [40]. In this study, the forage fed first in group C1 contained a high crude fiber content, which can stimulate intestinal peristalsis, increase fermentation efficiency, and promote volatile fatty acid production. The increased abundance of firmicutes can encourage the production of volatile fatty acids [38,41]. This is consistent with the results of the present study.

It has been reported that fecal matter can easily be collected and used as suitable samples to generate biomarkers to evaluate the gut microbiota [42]. Thus, fecal samples in this study were selected to evaluate the effect of the concentrate feeding sequence on the gut microbiota. Furthermore, 16S rRNA high-throughput sequencing technology can well reveal the fecal microbial diversity of weaned donkeys. In this trial, 1,226,185 optimized sequences with an average length of 417 bp were obtained from 15 samples sequenced by the Illumina MiSeq platform, and 4686 OTUs were obtained by co-clustering. This study found that coverage higher than 97% indicated adequate sampling of the sequencing samples. In this experiment, the range of all groups was higher than 0.99, indicating that the sequencing results can genuinely reflect the species and structural diversity of the fecal bacterial community in weaned donkeys. The Shannon index and Simpson index can reflect the variety of fecal flora. The more significant the Shannon index, the higher the community diversity. The lower the Simpson index, the higher the community diversity. The Chao1 index can reflect the richness of fecal flora, and the larger the value, the higher the community richness. In this study, different feeding methods affected the diversity and richness of the fecal bacteria.

The animal gut is one of the most densely inhabited microbial habitats and represents a highly specialized internal ecosystem. The commensal gut microbiota may impact the host’s health, immunity, metabolic capacity, and growth performance [43,44]. Most studies focus on the diversity of microorganisms in the digestive system of humans and ruminant animals. As a non-ruminant animal, the donkey has received few reports. Weaned 6-month-old male donkeys were selected for high-throughput 16S rRNA sequencing in this study to assess the diversity and abundance of fecal bacteria under various feeding methods. Studies have shown that different feeding methods could regulate the fecal microbial community at the phylum level. The dominant phyla in this study were Firmicutes and Bacteroidota, which represented more than 75% of the fecal microbial population in the weaned donkeys, agreeing with the observations from earlier studies on microbial communities of *Equus* animals. For instance, Firmicutes and Bacteroidetes are both primary in the gut of Dezhou donkeys, according to the study of Liu et al. [45]. Likewise, Zhang et al. reported that Firmicutes and Bacteroidetes predominated in digestive and mucosal associated microbiota at different intestinal sites in donkeys [46]. Su et al. discovered that Firmicutes (55.01%) and Bacteroidetes (24.76%) had the highest abundances in horses [47]. Additionally, Zhao et al. showed that Firmicutes and Bacteroidetes were the most prevalent and numerous phyla in horse fecal samples [48]. Gut-dwelling bacteria can be generically categorized according to their functions as proteolytic, lactate-using, glycolytic, and cellulolytic bacteria, the latter of which mainly include Firmicutes and Bacteroidetes [49,50] In this study, the relative abundance of Bacteroidota varied significantly among the three groups, with the relative abundance of Firmicutes in group C1 being significantly higher than that in the other groups, which may be associated with the feeding method. The ratio of Firmicutes to Bacteroidota in the gut affects the absorption of nutrients in the feed. Studies have shown that feeding high-energy (HE) diets to donkeys significantly decreased the ratio of Firmicutes to Bacteroidetes (F/B), with an increased richness of Bacteroidetes, which may be an important factor in improving growth performance [51], which is inconsistent with the results of this experiment. We hypothesized that this may be due to the small sample size in the previous study and the difference in species. To summarize, we discovered that group C3 had the best growth performance, with increasing abundances of Bacteroidetes and lower abundances of firmicutes. Spirochaetota can degrade cellulose, hemicellulose, and pectin, which has an important effect on the conversion of plant fiber materials into VFA [52]. In this trial, the relative abundance of Spirochaetota in group C3 was significantly higher than that in the other groups, indicating that total mixed ration feeding might facilitate the degradation of fibers by gut microorganisms, and the Fibrobacterota closely related to fiber degradation was also higher than that in the other groups. This suggests that variations in the VFA concentration may be related to the degradation degree of fiber and non-fiber substances by different microorganisms. Liu et al. [53] discovered that the Unidentified-Spirochaetaceae and the Anaerovibrio dominated the cecum flora of the Dezhou donkeys. In this experiment, the Rikenellaceae-RC9-gut-group and Unidentified-F082 were the dominant genera in the intestines of the weaned donkeys, which is inconsistent with the findings of earlier studies and may be due to differences in the species, age, dietary structure, and feeding management. According to previous studies, gut microbe composition and diversity are dynamic and can be influenced by many factors, such as diet [54], gender [55], age [56], and environment [57]. In this study, we observed significant differences in the microbial composition among the various treatment groups. Prior research has demonstrated that the effects of diet on gut microbes are similar in rabbits [58] and humans [59], with higher Bacteroides abundance and lower firmicutes abundance, and various nutritional compositions may be the cause of the diverse biodiversities. In addition, animals may receive different nutrients at different stages of growth, which may lead to differences in the gut flora. The gut microbial structure of weaned donkeys was significantly different under different feeding patterns, with the gut microbial composition of group C1 being significantly different from that of the other groups. It has been hypothesized that growth performance in weaned donkeys is correlated with changes in the microbial composition of the gut. However, the specific relationship between the two needs to be further verified.

## 5. Conclusions

In this study, it was shown that the concentrate feeding sequence can affect the growth performance of weaned donkeys by changing the gut microbial composition and proportion and nutrient digestibility. The TMR method significantly increased the average daily weight gain of weaned donkeys for optimal growth performance. These findings provide more insight into the gut microbes of weaned donkeys, but how these microbiota interact to affect growth performance needs further study.

## Figures and Tables

**Figure 1 animals-13-02893-f001:**
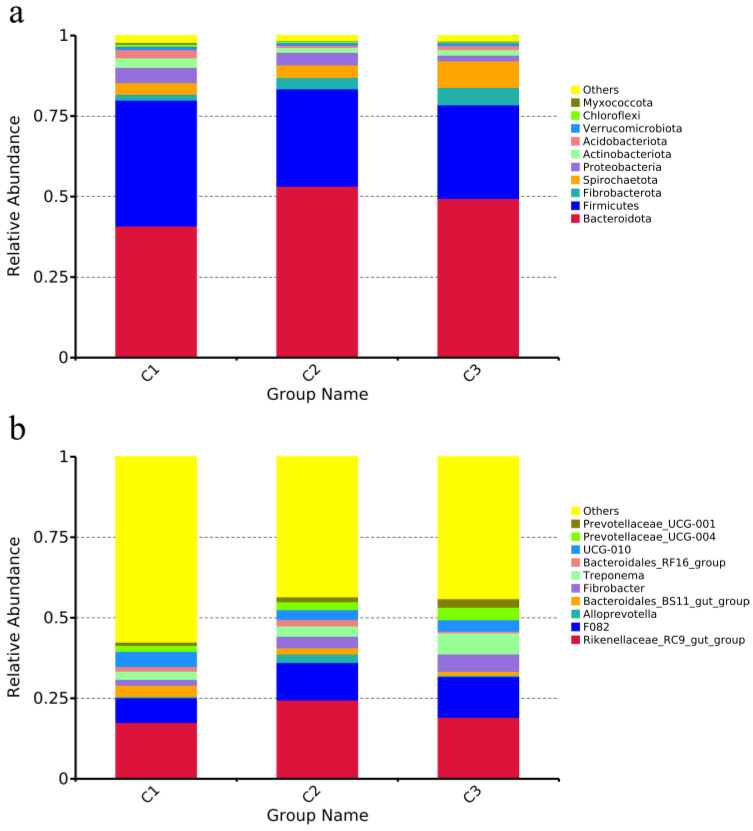
Species relative abundance histogram. (**a**) Relative abundance of fecal microbial species in weaned donkeys at the phylum level (top 10); (**b**) relative abundance of fecal microbial species of weaned donkeys at the genus level (top 10).

**Figure 2 animals-13-02893-f002:**
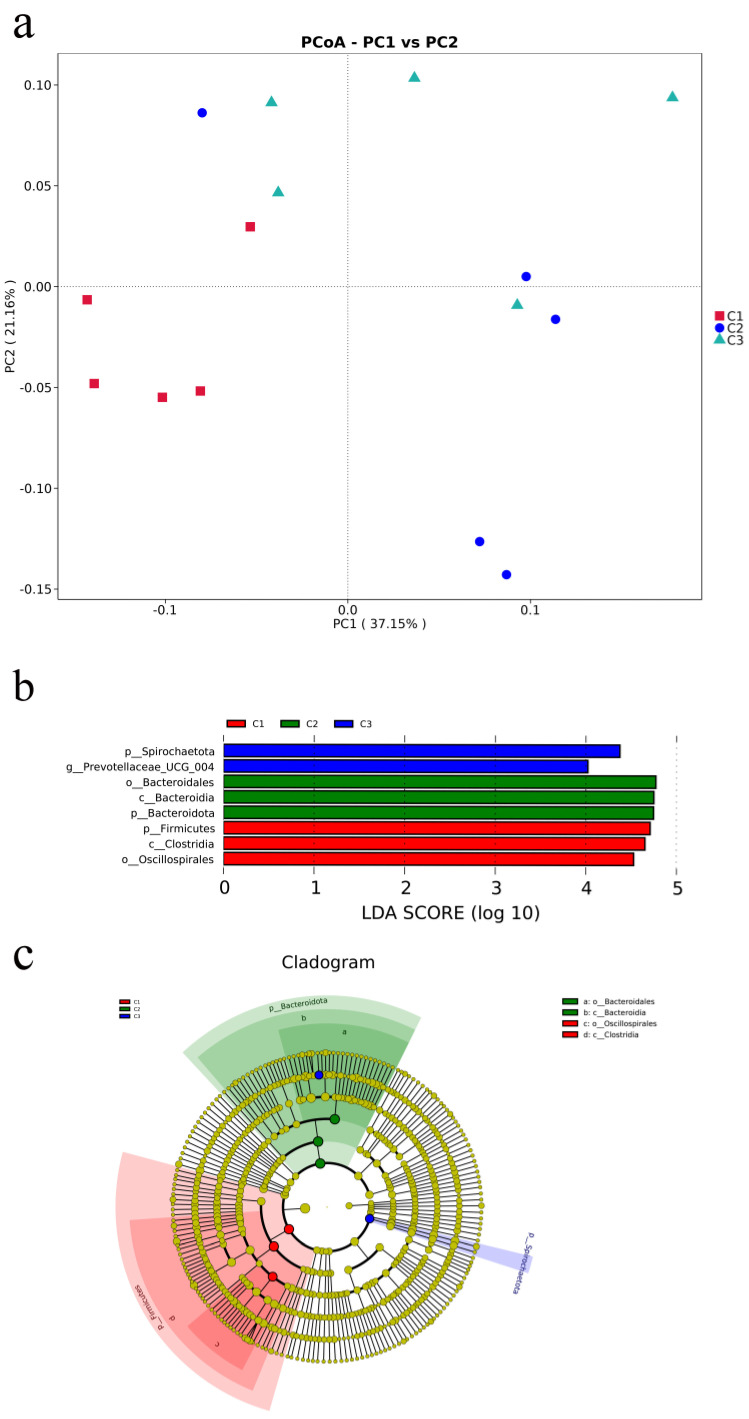
Microbial community analysis of fecal microbiota. (**a**) Principal coordinates analysis (PCoA) was carried out based on weighted−unifrac distances with *p*−values of anosim to highlight the differences in the fecal microbiota communities of the three groups. (**b**) Linear discriminant analysis (LDA) value−distributed histogram, only taxa meeting an LDA significant threshold > 4 are shown. (**c**) Cladogram constructed to visualize the microbial community relative abundance data of the three groups.

**Figure 3 animals-13-02893-f003:**
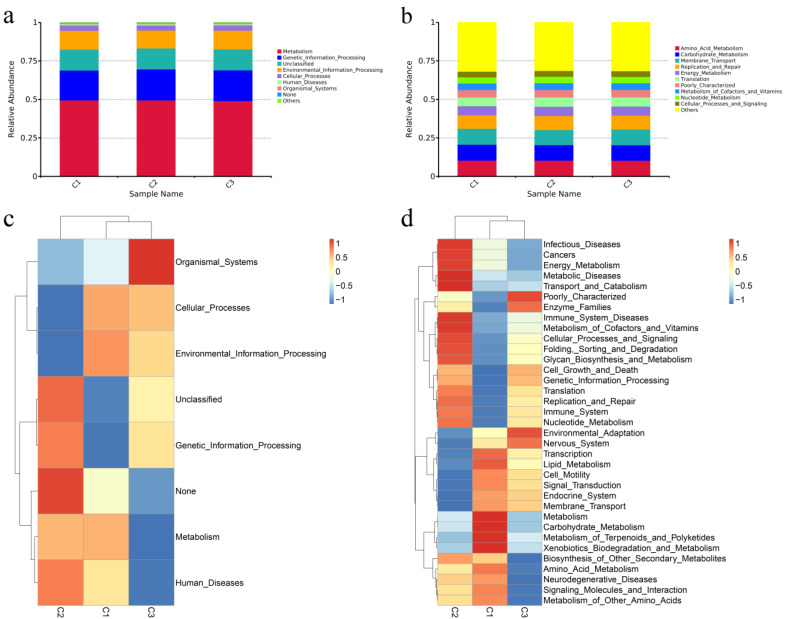
Functional predictions of the intestinal microbiota using PICRUSt. (**a**,**b**) Relative enrichment of KEGG level 1 and level 2 pathways in intestinal microbiota of donkeys with different concentrate feeding sequences. (**c**,**d**) Corresponding heatmaps for KEGG level 1 pathways and KEGG level 2 pathways.

**Table 1 animals-13-02893-t001:** Composition and nutrient levels of the concentrate (air-dry basis)/%.

Items	Content
Ingredients	
Corn grain (GB2 grade)	35.43
Wheat flour (NY/T1 grade)	20.00
Corn germ meal	25.00
Cornstarch residue	10.00
Mineral meal	1.97
Calcium hydrogen phosphate	1.14
Salt	0.60
Lysine	0.85
Threonine	0.10
Flavoring agent	0.02
Magnesium sulfate	0.20
Microminerals for donkey ^1^	0.20
Donkey vitamin ^2^	0.15
Soybean meal (GB1 grade)	4.34
Total	100
Nutrient levels ^3^	
Crude protein	16.03
Crude fiber	13.28
Ether extract	2.08
Acid detergent fiber	23.08
Neutral detergent fiber	16.93

^1^ Microminerals for donkey: Fe—80 mg, Mn—20 mg, Zn—88 mg, Cu—11 mg, Se—0.18 mg, I—0.82 mg. ^2^ Donkey vitamin: vitamin A—5000 IU, vitamin D—145.8 mg, vitamin E—2.2 mg, vitamin K—0.5 mg. ^3^ Nutrition levels were measured.

**Table 2 animals-13-02893-t002:** Nutrient level of peanut vine (air-dry basis)/%.

Nutrient Levels	Content
Crude protein	6.62
Crude fiber	67.24
Ether extract	0.39
Acid detergent fiber	79.51
Neutral detergent fiber	82.02

**Table 3 animals-13-02893-t003:** Effects of different concentrate feeding sequences on growth performance.

Indicators	C1	C2	C3	*p* Value
Initial BW, kg	117.00 ± 5.56	117.20 ± 4.44	117.20 ± 5.30	0.999
Final BW, kg	127.60 ± 5.92	131.50 ± 5.13	132.90 ± 6.33	0.803
ADFI, kg/day	2.69 ± 0.02 ^b^	2.79 ± 0.02 ^a^	2.76 ± 0.02 ^a^	0.022
ADG, kg/day	0.30 ± 0.02 ^b^	0.41 ± 0.04 ^a^	0.45 ± 0.03 ^a^	0.003
F/G	9.30 ± 0.69 ^a^	7.28 ± 0.59 ^b^	6.33 ± 0.34 ^b^	0.003

Note: The data are the means of five replicates per treatment (*n* = 5). The values with different letter superscripts mean significant differences (*p* < 0.05). BW = body weight; ADFI = average daily feed intake; ADG = average daily gain; F/G = feed intake/gain.

**Table 4 animals-13-02893-t004:** Effects of different concentrate feeding sequences on apparent digestibility.

Indicators	C1	C2	C3	*p* Value
CP	73.05 ± 0.008 ^b^	76.82 ± 0.009 ^a^	77.72 ± 0.005 ^a^	0.002
EE	35.40 ± 0.019 ^b^	50.41 ± 0.019 ^a^	46.63 ± 0.026 ^a^	0.002
CF	63.77 ± 0.539	63.07 ± 0.555	64.00 ± 0.947	0.637
NDF	67.09 ± 0.827	66.33 ± 0.858	67.83 ± 0.675	0.434
ADF	59.03 ± 0.490	58.58 ± 0.275	60.36 ± 0.712	0.081

Note: The data are the means of five replicates per treatment (*n* = 5). The values with different letter superscripts mean significant different (*p* < 0.05). CP = crude protein; EE = crude extract; CF = crude fiber; NDF = neutral detergent fiber; ADF = acid detergent fiber.

**Table 5 animals-13-02893-t005:** Effects of different concentrate feeding sequences on fecal VFA (μg/mL) levels.

Indicators	C1	C2	C3	*p* Value
Acetic acid	220.97 ± 15.48 ^a^	153.74 ± 16.38 ^b^	133.94 ± 20.76 ^b^	0.017
Propionic acid	43.65 ± 6.85	38.46 ± 8.50	36.20 ± 5.30	0.750
Butyric acid	12.45 ± 3.43	10.14 ± 2.33	9.80 ± 2.29	0.764
Isobutyric acid	8.36 ± 0.90 ^a^	4.90 ± 0.18 ^b^	4.11 ± 0.35 ^b^	0.001
Valeric acid	4.78 ± 0.50 ^a^	2.48 ± 0.55 ^b^	1.85 ± 0.23 ^b^	0.003
Isovaleric acid	5.85 ± 0.78 ^a^	3.93 ± 0.50 ^b^	2.73 ± 0.44 ^b^	0.014
Caproic acid	0.19 ± 0.03 ^a^	0.10 ± 0.04 ^ab^	0.05 ± 0.01 ^b^	0.032
Total VFA	296.26 ± 19.64 ^a^	213.74 ± 26.04 ^b^	188.67 ± 23.80 ^b^	0.024

Note: The data are the means of five replicates per treatment (*n* = 5). The values with different letter superscripts mean significant differences (*p* < 0.05). VFA = volatile fatty acids.

**Table 6 animals-13-02893-t006:** Characteristics of amplicon libraries.

Characteristic	C1	C2	C3	SEM	*p* Value
Sequences	80,677.00	79,425.00	82,771.75	624.090	0.071
OTUs	1614.25 ^a^	1245.00 ^b^	1283.50 ^b^	67.530	0.028
Chao1 index	1617.40 ^a^	1246.41 ^b^	1286.06 ^b^	67.520	0.027
Shannon index	9.52 ^a^	8.51 ^b^	8.88 ^ab^	0.180	0.040
Simpson index	1.00	0.99	0.99	0.002	0.305

Note: The values with different letter superscripts mean significant differences (*p* < 0.05). OUT = operational taxonomic unit; SEM = standard error of the mean.

## Data Availability

All data generated or analyzed during this study are included in this paper.

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
