# Peer review of "Effects of Concentrate Feeding Sequence on Growth Performance, Nutrient Digestibility, VFA Production, and Fecal Microbiota of Weaned Donkeys"

_animals, 2023, doi:10.3390/ani13182893_

Round 1
Reviewer 1 Report
Dear authors,
Many thanks for this piece of work which I read with great interest. The title is pertinent to the text but the abstract is poorly developed. In particular concepts are sometimes disconnected from each other. My suggestion is to start directly with objectives.
Terms used are sometimes colloquial. Instead, prefer more technical terms like feeding practice and not feeding mode (just an example).
Please, also report how you assess the nutritional status of the donkey in general, and then focus on the breed and state why it is important to assess it. For your convenience, I suggest you to refer to Cappai et al., 2013, IJAS doi:10.4081/ijas.2013.e29
Please, tailor discussion to results. It is extremely oriented to comment microbiological outcomes, almost exclusively. But your paper deals with several other parameters. So, please, don't miss any and be more focused.
Conclusion must report succinctly the importance of results. If you improve the Introduction and clearly state the hypothesis, your conclusion will meet the question with new answers, coming from your results.
Tables and graphs are fine.
Thank you.
Reviewer 2 Report
The manuscript investigates the impact of three different feeding patterns, including fiber-to-concentrate (FC), concentrate-to-fiber (CF), and total mixed ration (TMR), on the growth performance, nutrient digestibility, VFA production, and gut microbiota of weaned donkeys. The findings of the study suggest that the TMR group had the best growth performance and that feeding patterns can change the composition of the gut microbe of weaned donkeys. The study's results could be useful for animal nutritionists and veterinarians to develop feeding strategies for weaned donkeys that can improve their growth performance and gut health. Overall, the subject itself is surely worthy of investigation. However, some points need to be addressed as follows:
1) Throughout the manuscript, Gut VFA or microbes should be replaced to be “fecal VFA or microbes”.
2) In the manuscript, one of the experimental groups was abbreviated as CF. This could cause confusion for readers, as CF is also an abbreviation for crude fiber (line 119). Therefore, I suggest that the authors revise and correct the abbreviation used for the experimental group to avoid confusion.
3) In the introduction section, the authors described the TMR method in detail. However, it is not clear why the feeding sequence of fiber then concentrate or concentrate then fiber was used in the FC and CF groups. This information would help readers understand the rationale behind the feeding patterns used in the study and provide a more comprehensive understanding of the study's results.
4) Line 84: please clarify the sex (male, female, or both) of the weaned donkeys.
5) Line 87: replace “wad” with “was”.
6) Line 95-96: “The CF group was fed forage for half an hour, then concentrated feed, and vice versa for the CF group.” The meaning is not clear, please rewrite it.
7) Throughout the manuscript, change “concentrate-to-fiber” to “concentrate-then-roughage” and also change “fiber-to-concentrate” to “roughage-then-concentrate” to describe the feeding sequence better.
8) Line 118: “Zou and Hassanat, F [16,17]” These references are secondary citations, not the original. The authors must use the original references for the method of evaluating digestibility.
9) The analysis methods of the concentrate (Table 1) should be mentioned in the M&Ms section.
10) Delete “This is a” from Tables 1 and 2.
11) Table 3: Why are small and capital letter superscripts used? And what is the difference between them? if no differences please use small letters only.
12) In the statistical analysis section, clarify the post hoc test you used.
13) Table 1: add the formulation of both Microminerals and Donkey vitamins in the table footnote.
In all Tables, describe the experimental groups and all abbreviations used in the table's footnotes. Describe also the experimental groups and the number of analyzed samples (n=?).
Minor editing of English language required.
Reviewer 3 Report
Manuscript is interesting and suggesting important considerations, when designing donkey's feed rations.
Yet, it suffers from some minor problems; I believe that most of them can easily be solved by revising the manuscript.
Simple summary:
· Lines 15-17: the sentence is long and not clear –please revise.
Abstract:
· When introducing shortcuts for the first time, the full name to which the shortcut stands for, should be written (for example: CP, EE).
· Line 39: the words "the difference genus" is unclear – please revise.
· Lines 39-42: the sentence is not clear - please revise.
Introduction:
· Lines 76-80: the sentence is too long and should be divided - please revise.
Material and methods:
· In general, section should be re-edited to ensure proper scientific writing and clarity regarding: procedures and reagents, which not always present (sulfuric acid, phosphoric acid and etc.).
· Line 107: the word centrifuged is missing before 200g – please revise.
· Line 123: centrifugation speed should be written in g and not in rpm.
· Details in section 2.4 are lacked – please add details regarding the mentioned procedures.
· Section 2.6: how was the statistical significance determined? Since the comparisons between more than 2 groups were made – you should have used Tukey HSD test post ANOVA. Please address this point in this subsection.
Results:
· All findings presented in the results, should be addressed, even if no change was observed between treatments. Moreover, Differences between groups should be presented in the text, even if they are not significant (for example: section 3.1, section 3.4.1, section 3.4.2).
Please revise the result section accordingly.
· Please use small letter for describing statistical significance in the tables. Currently, there is a mix of small and capital letters – please revise.
· Section 3.4.2: please add the exact relative abundance of the bacteria next to significance. It would be helpful for the reader to assess the differences. In addition, you should mention that around 50% of the bacteria species found (figure b) were undefined (others).
· Line 241: please insure the text reference to figure 3c is correct – I think not.
· Section 3.4.4 is not clear which makes it hard to understand - please revise.
Discussion and conclusions:
· All together, the discussion is well written.
· Line 254-255: the sentence "There 254 were more studies on ruminants", seems to be out of context –please revise.
· Line 259: to be accurate, the differences found are statistically different when regarding the FC group comparing the TMR group (but not the CF group) – please revise.
· Lines 267-269: sentence is not clear - please revise.
English and editing:
· Line 16 – please complete the word can after the word feeding.
· Line 22: the words as a result are not suitable – please revise.
· Line 35: please replace the comma after the word for example to colon (:).
· Line 87: add point after the word province and fix the word wad to was.
· Line 351: please replace the point after factors to comma.
Round 2
Reviewer 1 Report
Dear authors,
Thank you for amends to your paper. I am still doubtful about some passages in the protocol. You meant to apply feeding practices (not really patterns!) to a single stomached, roughage and and feeds high in structural fibre adapted herbivore, namely the donkey.
It is still astonishing to me that you selected just weaned donkeys to be fed on high concentrate diets (how long?) and no evaluation on the digestive function (feces quality, clinical health status) were conducted. No nutritional assessment is reported. Technical terms are sometimes wrong and discussion (likewise introduction) is build up on findings reported for calves, which cannot be assumed to be interpreted as valid as in donkeys. Not even from a comparative approach in the light of the different nutrition physiology between the two animal species and the outcome you meant to achieve for intensively raised donkeys.
Sorry but I am not in favour of publishing the paper.
Technical terms need to be corrected and appropriate to the animal species.
Reviewer 2 Report
All comments have been addressed.
